# Targeted Therapy for a Rare *PDGFRB*-Rearranged Myeloproliferative Neoplasm: A Case Report

**DOI:** 10.3390/ijms27020656

**Published:** 2026-01-08

**Authors:** Cosimo Barbato, Vito A. Lasorsa, Francesco Grimaldi, Santa Errichiello, Ida Pisano, Maurizio Capuozzo, Mariangela Capone, Viviana Izzo, Fabrizio Quarantelli, Alessandra Potenza, Roberta Visconti, Alessandra Galdiero, Angelo Zanniti, Ciro Del Prete, Teresa Femiano, Giuseppina Esposito, Novella Pugliese, Roberta Russo, Mario Capasso, Barbara Izzo

**Affiliations:** 1CEINGE Biotecnologie Avanzate Franco Salvatore, 80145 Naples, Italy; lasorsa.alessandro@gmail.com (V.A.L.); errichiello@ceinge.unina.it (S.E.); capuozzo@ceinge.unina.it (M.C.); caponem@ceinge.unina.it (M.C.); quarantelli@ceinge.unina.it (F.Q.); potenza@ceinge.unina.it (A.P.); viscontir@ceinge.unina.it (R.V.); galdiero@ceinge.unina.it (A.G.); zanniti@ceinge.unina.it (A.Z.); delpreteci@ceinge.unina.it (C.D.P.); femiano@ceinge.unina.it (T.F.); espositogiu@ceinge.unina.it (G.E.); roberta.russo@unina.it (R.R.); mario.capasso@unina.it (M.C.); 2Department of Molecular Medicine and Medical Biotechnology, University of Naples “Federico II”, 80131 Naples, Italy; 3Hematology Section, Department of Clinical Medicine and Surgery, University of Naples “Federico II”, 80138 Naples, Italy; francesco.grimaldi1@unina.it (F.G.); novypugliese@yahoo.it (N.P.); 4UOSD Medical Genetics Laboratory, ASL Napoli 1 Centro, 80141 Naples, Italy; ida.pisano@aslnapoli1centro.it; 5UOC of Clinical Pharmacology, University Hospital ‘San Giovanni di Dio e Ruggi d’Aragona’, 84131 Salerno, Italy; vizzo@unisa.it; 6Department of Medicine, Surgery and Dentistry “Scuola Medica Salernitana”, University of Salerno, 84084 Fisciano, Italy

**Keywords:** myeloproliferative neoplasms (MPNs), *PDGFRB* gene, *CCDC88C* gene, cytogenetics, whole genome sequencing (WGS), RNA sequencing (RNAseq), imatinib

## Abstract

Myeloproliferative neoplasms (MPNs) are a heterogeneous group of diseases originating from hematopoietic stem cell transformation, characterized by the clonal proliferation of hematopoietic progenitors. A specific subset includes myeloid/lymphoid neoplasms with eosinophilia and tyrosine kinase (TK) gene fusions, particularly involving *PDGFR A* or *B*, which are sensitive to TK inhibitor treatment. We report a case of a 21-year-old patient with a myeloproliferative/myelodysplastic neoplasm, presenting with hyperleukocytosis, anemia, thrombocytopenia, and elevated LDH. The peripheral blood smear showed hypogranular neutrophils, eosinophils, basophils, and myeloid precursors. The absence of *BCR::ABL1* and mutations in *JAK2*, *CALR*, and *MPL* excluded common MPNs. Cytogenetic analysis revealed a rearrangement between chromosomes 5 and 14. FISH analysis confirmed an inverted insertion from chromosome 5 to chromosome 14, involving the *PDGFRB* gene. WGS and RNAseq identified a fusion between *PDGFRB* and *CCDC88C*, causing the constitutive activation of *PDGFRB*. The fusion gene was confirmed by sequencing. This allowed for targeted therapy with a tyrosine kinase inhibitor (TKI), leading to molecular remission monitored by RT-qPCR. This case highlights how a multidisciplinary approach can identify atypical transcripts in MPN, guiding targeted therapy with TK inhibitors, thus resulting in effective treatment and molecular remission.

## 1. Introduction

The *PDGFRB* gene (5q32) codes for the platelet-derived growth factor receptor-beta, which is characterized by five extracellular immunoglobulin-like domains and a split intracellular kinase domain. PDGFRB binds preferentially to members of the PDGF ligand family, particularly PDGF-B and PDGF-D. Ligand engagement triggers receptor dimerization, followed by tyrosine autophosphorylation and the activation of key intracellular signaling pathways. These include the Ras/MAPK, PI3K/Akt, and PLCγ cascades, which collectively promote cell proliferation, differentiation, survival, and migration processes that are essential during embryonic development and for maintaining tissue homeostasis in adult organisms. The excessive activation of these receptors is implicated in various malignancies as well as in disorders characterized by uncontrolled cell proliferation [1,2]. Gene fusions following chromosomal translocations involving the *PDGFRB* gene identify a specific subgroup of hematological malignancies defined as myeloid/lymphoid neoplasms with eosinophilia and tyrosine kinase gene fusions (MLN-TK) by the latest WHO/ICC classifications [3,4]. These malignancies are extremely rare, accounting for less than 1% of all myeloid neoplasms. The true incidence is difficult to estimate due to underdiagnosis and historical misclassification; they can occur at any age, but are most frequently diagnosed in young to middle-aged adults, with a reported median age ranging between 30 and 50 years, with a slight male predominance reported [5,6]. They frequently share clinical features with myelodysplastic/myeloproliferative neoplasms (MDS/MPNs), are commonly associated with peripheral eosinophilia, and show high sensitivity to tyrosine kinase inhibitors (TKI) [7,8]. While hypomethylating agents and hydroxyurea are commonly employed in high-risk MDS/MPN patients with a low blast count in the absence of actionable molecular targets, patients harboring *PDGFRB* rearrangements, even in this clinical setting, typically achieve rapid and durable molecular remissions with low-dose tyrosine kinase inhibitors. Cytogenetic and molecular monitoring have consistently demonstrated that low-dose imatinib (100–200 mg daily) is sufficient to induce and maintain long-lasting molecular responses [9].

## 2. Results

### 2.1. Case Report

We herein describe a case of a 21-year-old male patient who initially presented to the emergency department with symptoms of fatigue and fever. A complete blood count revealed hyperleukocytosis with a white blood cell (WBC) count of 115,000/mm^3^, hemoglobin (Hb) of 9.7 g/dL, thrombocytopenia with a platelet count (PLT) of 116,000/mm^3^, and elevated lactate dehydrogenase (LDH) levels of 850 IU/mL. Based on these findings, the patient was urgently transferred to AOU Federico II Hematology Division for further investigations. The peripheral blood smear showed 35% hypogranulated neutrophils, along with immature cells, including promyelocytes (4%), myelocytes (25%), and metamyelocytes (10%). Additionally, eosinophilia (7% of WBC—7170/mmc) and basophilia (2% of WBC—2048/mmc) were observed. The abdominal ultrasound study revealed a mild splenomegaly (15 cm longitudinal diameter). Cytoreduction with hydroxyurea was then started and a trephine biopsy performed, which revealed an increased bone marrow cellularity and hyperplasia of megakaryocyte and granulopoietic precursors, with increased eosinophilic differentiation. No excess of blast cells was detected. Flow cytometry confirmed the absence of blast cells, and the presence of a granular hyperplasia (89%) with abnormalities in maturation. Peripheral blood molecular analysis by PCR excluded *BCR::ABL1* rearrangement, mutations in the *JAK2* gene, and overexpression of the *WT1* gene. Mutational analysis was extended by using a next-generation sequencing (NGS) panel of 30 myeloid-targeted genes (Table 1). NGS analysis did not allow the identification of any pathogenic variants in the genes analyzed.

### 2.2. Cytogenetic Analysis

Cytogenetic evaluation revealed in 90% of the metaphases a rearrangement involving one copy of chromosome 5 and one copy of chromosome 14 (Figure 1A). The chromosome rearrangement consisted of a balanced translocation involving the long arm of a chromosome 5 at position q12 and the long arm of a chromosome 14 at position q32. To better understand the nature of this rearrangement, a FISH with whole chromosome painting probes was performed, revealing that the aberration consisted of an insertion of a large segment of the long arm of a chromosome 5 into the long arm of a chromosome 14 (Figure 1B). To better define the potential chromosome breakpoints and chromosomal regions involved in the anomaly, a FISH with locus-specific probes was set up. For chromosome 5, a mixture of three probes was used: XL 5q31/5q33/5p15 Metasystems FISH probes. The first hybridizing on the short arm at the level of the p15 chromosomal region and labeled with the fluorophore aqua was used as a hybridization control. The other two probes hybridize on the chromosomal region 5q31 (labeled with the orange fluorophore) and region 5q33 (labeled with the green fluorophore). As shown in Figure 1, the FISH assay displayed normal signals on the intact copy of chromosome 5, whereas for the pathological chromosome 5, the loss of the 5q31 region was observed. This region was found to be relocated on the derivative chromosome 14 in a region much further upstream than anticipated. Noteworthy, in a simple insertion, this marker would be typically located in the terminal portion of the derivative chromosome 14. However, in this case, the marker was found in a diametrically opposite position to what would be expected. This condition suggests that the chromosome segment was inserted in an inverted orientation within the recipient chromosome. Overall, this experiment confirmed the presence of the insertion and further revealed that the patient also had an inversion (Figure 1C).

Based on this result, we hypothesized that the involvement of the *IGH* gene as the insertion point on chromosome 14 is located near its locus. To verify this, a FISH assay with break-apart (BA) probes for the *IGH* gene was performed, which yielded a negative result, thus excluding the involvement of the *IGH* gene (Figure 1D). Subsequently, the XL 5q32 *PDGFRB* BA Metasystems FISH probes were used for the *PDGFRB* gene, located on chromosome 5q32 and occasionally involved in hematologic malignancies. The FISH assay revealed the relocalization of the 3′ region of the *PDGFRB* gene on chromosome 14 at position q32 (Figure 1E), which confirmed the *PDGFRB* rearrangement without, however, identifying the specific fusion partner.

### 2.3. Whole Genome Sequencing and RNA Sequencing Analysis

At this point, it was decided to proceed with the analysis of somatic variants by whole genome sequencing (WGS) on DNA isolated from the bone marrow matched with germline DNA from peripheral blood as the control tissue. This experiment was performed to validate the results obtained from the FISH experiments and to better understand the nature of the inverted insertion. It is well known that apparently balanced events, as observed at the chromosomal level, may hide deletions or duplications at the breakpoints that are cryptic at the karyotype level. The results of the somatic copy number alteration (CNA) analysis demonstrated the presence of two regions of loss (with CN = 1) both mapping at 5q11 (Figure 2). The first copy number alteration was approximately 1.9 Mb, contained 17 genes, and was classified as pathogenic according to the American College of Medical Genetics (ACMG) guidelines. The second copy number alteration, located roughly 7 Mb downstream, was about 150 Kb and was classified as a variant of unknown significance (VUS) as it did not contain any gene. Moreover, in the somatic structural variant (SV) analysis, breakpoints were found to support an inverted insertion event of a large segment of chromosome 5 from 5q11 to 5q32 (about 92 Mb containing 694 genes) on chromosome 14q32 (Figure 2). To better characterize the *PDGFRB* fusion partner and assess its expression levels, an RNA sequencing experiment (RNAseq) was carried out. The bioinformatics analysis of the transcriptome and the search for fusion genes confirmed and added details to the data obtained by WGS. Specifically, three fusion events were identified: the first between the *PDGFRB* gene (on chromosome 5) and *CCDC88C* (on chromosome 14); the second fusion involved the genes *SKIV2L2* and *C14orf159* (on chromosomes 5 and 14, respectively); and the third involved the genes *SKIV2L2* and *GPBP1* (both on chromosome 5). The first two fusion events supported the inverted insertion of chromosome 5 on chromosome 14 whereas the third fusion was a consequence of the pathogenic CN loss involving *SKIV2L2* and *GPBP1* (both on chromosome 5). Additional investigations showed that only the *PDGFRB::CCDC88C* fusion gene (Figure 3A) retained a complete open reading frame, potentially preserving the integrity of the original protein domains. The resulting fusion protein, comprising 1028 amino acids, is characterized by the hook-type domain of the CCDC88C protein and the kinase domain of the PDGFRB protein (Figure 3B). Finally, RNAseq analysis highlighted that the fusion messenger RNA exhibited expression levels similar to those of *CCDC88C*, from which it has inherited the regulatory region (Figure 3C). Considering that the *PDGFRB::CCDC88C* fusion identified by our analysis is able to dimerize, and that the fusion product could lead to the constitutive activation of the kinase function of PDGFRB [10], a final diagnosis of myeloid/lymphoid neoplasms with eosinophilia and tyrosine kinase gene fusion according to WHO 2022 [3] was made, and the patient was started on low-dose imatinib (200 mg od), accordingly. Based on the fusion gene sequence predicted by the RNAseq analysis, a pair of primers was designed (Table 2) to selectively amplify the cDNA of the fusion gene at the diagnosis by qualitative RT-PCR and a probe (Table 2) was designed to follow the molecular response by RT-qPCR in the bone marrow of the patient during the targeted therapy.

## 3. Discussion and Conclusions

Myeloid/lymphoid neoplasms with *PDGFRB* rearrangements represent a biologically and clinically distinct subgroup of myeloid disorders, characterized by the constitutive activation of tyrosine kinase signaling and the remarkable sensitivity to tyrosine kinase inhibitors. Although the cytogenetic hallmark is often an apparently balanced rearrangement involving chromosome 5q, these events can mask complex structural abnormalities, as demonstrated in our case by the presence of an inverted insertion and cryptic copy number losses revealed only through whole genome sequencing. The identification of the rare *CCDC88C::PDGFRB* fusion further highlights the molecular heterogeneity of *PDGFRB*-associated neoplasms. *CCDC88C* encodes a protein involved in cytoskeletal organization and signal transduction, and when fused to *PDGFRB*, it provides a dimerization domain that leads to ligand-independent kinase activation. This mechanism explains the exquisite sensitivity of these neoplasms to low-dose imatinib, which has been consistently associated with rapid hematologic and durable molecular responses. From a clinical standpoint, this case emphasizes the importance of a multidisciplinary diagnostic approach, integrating morphology, cytogenetics, FISH, next-generation sequencing, and transcriptomic analysis. In young patients presenting with atypical myeloproliferative features, eosinophilia, and an absence of canonical driver mutations (*JAK2*, *CALR*, *MPL*, *BCR::ABL1*), an early search for targetable gene fusions is essential, as it can dramatically alter both prognosis and therapeutic strategy. In fact, younger patients, such as the one described in this report, can usually tolerate long-term tyrosine kinase inhibitor therapy and benefit from sustained molecular monitoring aimed at maintaining deep remission. More in general, the availability of targeted therapy allows for personalized treatment approaches across different age groups, significantly improving quality of life and long-term outcomes. Importantly, *PDGFRB*-rearranged neoplasms differ substantially from other MDS/MPN overlap syndromes in terms of treatment and outcome. While conventional cytoreductive agents may temporarily control leukocytosis, they do not address the underlying molecular driver. In contrast, targeted therapy with imatinib induces deep molecular remissions and can prevent disease progression, transformation, and organ damage. In conclusion, this timely, molecularly driven approach resulted in a rapid and sustained response, underscoring the importance of the early detection of *PDGFRB* rearrangements for optimizing patient outcomes. Our patient achieved a complete molecular remission with blood count and spleen size normalization on low-dose imatinib (200 mg/day), confirming previous observations that minimal effective dosing is often sufficient in this setting.

## Figures and Tables

**Figure 1 ijms-27-00656-f001:**
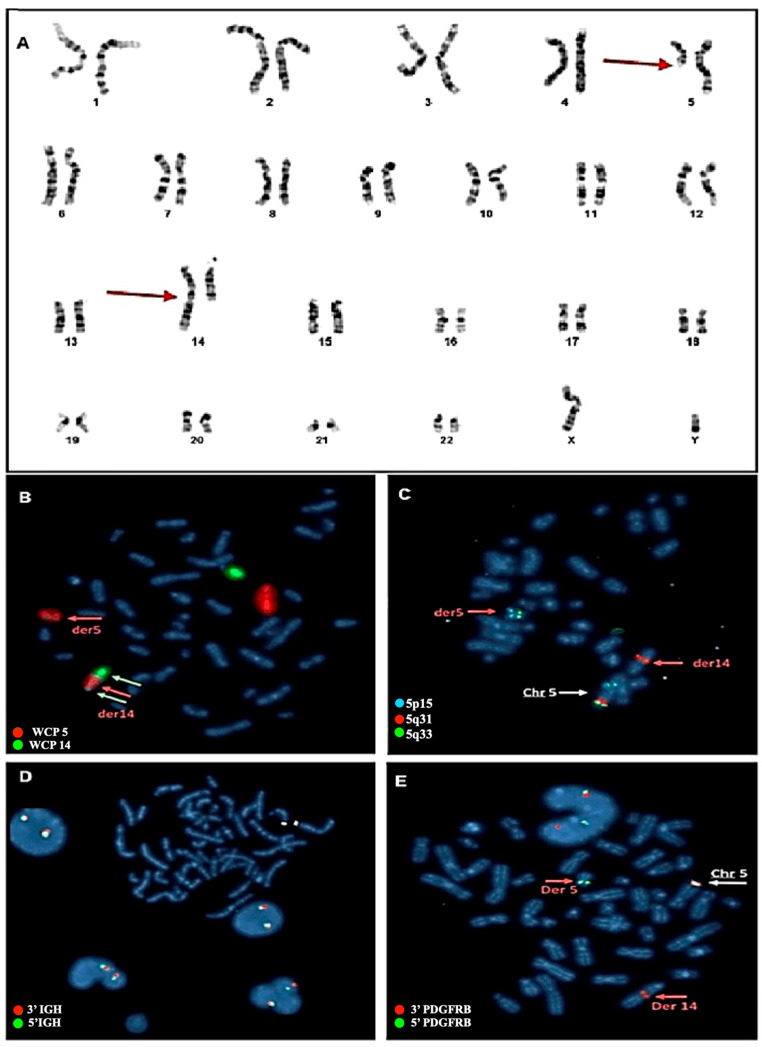
(**A**) Karyotype from bone marrow blood and molecular cytogenetic analysis with Metasystem probes. The red arrows indicate the rearrangement involving chromosome 5 and chromosome 14. (**B**) FISH with whole chromosome painting (WPC) probes for chromosomes 5 and 14. Probes that hybridize with chromosome 14 are marked in green, and probes hybridizing with chromosome 5 are marked with orange fluorophore. The image obtained shows how the portion of chromosome 5 (red arrow) goes inside chromosome 14 (white arrows), thus defining the rearrangement as an insertion and not a translocation. (**C**) FISH with LSI probes. One probe hybridizing to the 5p15 (control region) is labeled with Aqua fluorophore and the other two probes hybridizing to the 5q31 and 5q33 regions are labeled with orange and green fluorophores, respectively. The image obtained shows how the 5q31 region fits invertedly within the derivative chromosome 14, while the 5q33 region remains conserved in the derivative chromosome 5. (**D**) FISH with break-apart probes for the *IGH* gene. As is evident from the image, the gene shows no rearrangement. The two green and orange signals are close, highlighting the integrity of the gene, which is not involved in the rearrangement. (**E**) FISH with break-apart probe for the *PDGFRB* gene. In the image produced, the involvement of the gene in the rearrangement can be seen, highlighting how the 3′ region of the gene (marked with orange fluorophore) has relocated to the central part of derivative chromosome 14. In contrast, the 5′ region of the gene (marked with green fluorophore) remains preserved in the derivative chromosome 5.

**Figure 2 ijms-27-00656-f002:**
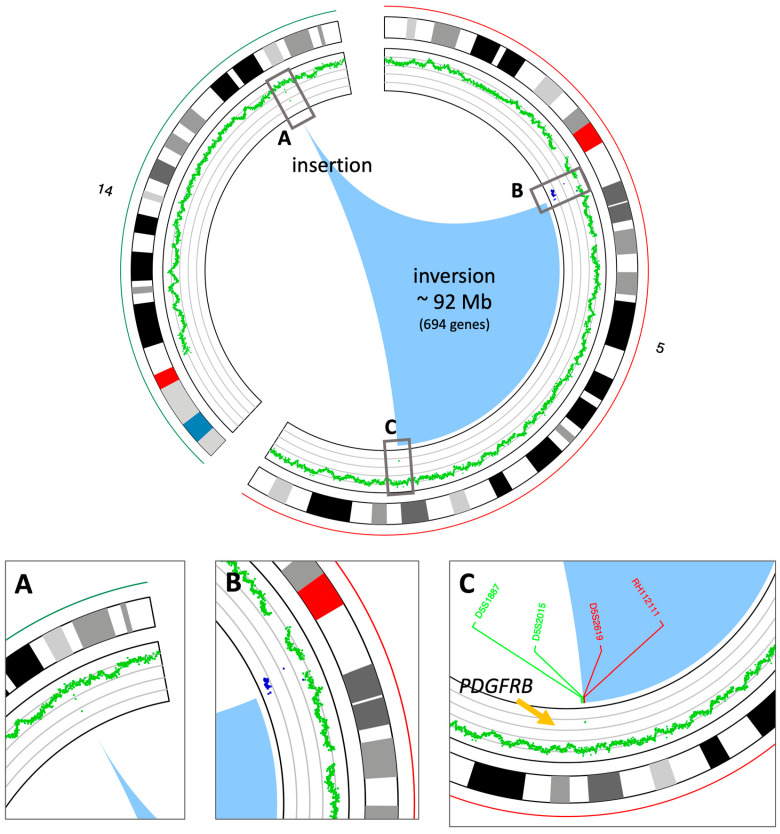
Circos view of the chromosomes involved in the inverted insertion. The circos plot shows, from outside to inside, the chromosome names and ideograms and the CN profile (green: CN = 2, blue: CN = 1); the arcs at the center (the light blue area) show the breakpoints on chromosome 5 and the insertion point on chromosome 14. (**A**) Zoom-in of the insertion point on chromosome 14. (**B**) Zoom-in of the upstream breakpoint on chromosome 5. CN losses are in blue. (**C**) Zoom-in of the downstream breakpoint on chromosome 5. The positions of the FISH probes shown in Figure 1 are also reported.

**Figure 3 ijms-27-00656-f003:**
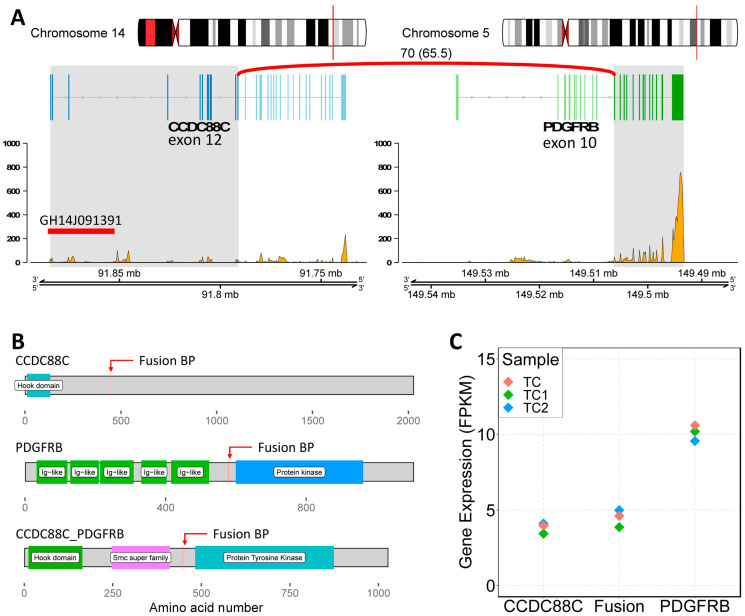
Characterization of the *PDGFRB::CCDC88C* fusion. (**A**) Genomic view of the fusion genes. On the left, *CCDC88C* locus on chromosome 14; on the right, *PDGFRB* locus on chromosome 5. The fusion event involved exons 1 to 12 of *CCDC88C* and from 10 to 22 of *PDGFRB* (gray highlighted areas). The plots report the normalized number of reads mapping on the mRNAs (from RNAseq data). GH14J091391: GeneHancer ID of the *CCDC88C* regulatory region. (**B**) Schematic view of the involved protein domains. (**C**) The plot shows the expression levels of the normal genes compared to the level of the fusion gene.

**Table 1 ijms-27-00656-t001:** NGS myeloid panel genes.

Gene	Exons Tested	Gene	Exons Tested
*WT1*	6–10	*PTPN11*	3, 7–13
*SETBP1*	4	*HRAS*	2, 3
*FLT3*	13–15, 20	*CALR*	9
*CBL*	8, 9	*IDH2*	4
*CEBPA*	All	*KRAS*	2, 3
*TP53*	2–11	*NRAS*	2, 3
*ETV6*	All	*SRSF2*	1
*MPL*	10	*SF3B1*	10–16
*BRAF*	15	*CSF3R*	All
*IDH1*	4	*EZH2*	All
*ASXL1*	9, 11, 12, 14	*KIT*	2, 8–11, 13, 17, 18
*JAK2*	All	*RUNX1*	All
*TET2*	All	*ABL1*	4–9
*U2AF1*	2, 6	*NPM1*	10, 11
*ZRSR2*	All	*DNMT3A*	All

**Table 2 ijms-27-00656-t002:** Oligonucleotide primers and probe used in RT-PCR and RT-qPCR.

Primer/Probe	Sequence	Tm (°C)
FW (Forward)	GAGATTGCACAGAAGCAGAG	47.8
RV (Reverse)	AGGATGATAAGGGAGATGATGG	47.6
Probe	FAM-5′ ACGCAGACTTGTCAGACGCCTTGCC 3′-TAMRA	-

## Data Availability

The data presented in this study are openly available in [Barbato, Cosimo (2026), “Data Set Case Report: Targeted therapy for a rare PDGFRB-rearranged myeloproliferative neoplasm”, Mendeley Data, V2, https://doi.org/10.17632/pys5v3ykhy.1 (accessed on 28 April 2025)].

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
