# Peer review of "Targeted Therapy for a Rare *PDGFRB*-Rearranged Myeloproliferative Neoplasm: A Case Report"

_ijms, 2026, doi:10.3390/ijms27020656_

Round 1
Reviewer 1 Report
Comments and Suggestions for Authors
The manuscript describes an interesting case with a rare PDGFRB rearrangement for which WGS and RNA sequencing was required to fully solve the type of rearrangement. However, there are some issues that should be discussed or adressed:
- a method section describing the experimental proceduces, probes used etc. ist completely missing
- labelling of figures is sometimes missing (e.g. Fig 1D), sometimes almost impossible to read (e.g. Fig 2C)
- discription of the rearrangement is confusing: while in line 107 a loss of 5q31 was reported, the rearrangement was overall described as balanced (line 95); it would be helpful to include the karyotype
- do the authors have any explanation for the high VAF observed for ZRSR2 and KIT? any additional CNVs or CN-LOH that could explain that?
- generally, although I agree that WGS and RNA seq was required to fully solve this case, it would not have been required for correct classification as myeloid/lymphoid neoplasm with PDGFRB rearrangement, FISH analysis would have been sufficient; this should at least be discussed and clarified what the benefit of WGS/RNA seq is
Comments on the Quality of English Language
the manuscript is sometimes very wordy and explanations are confusing
Author Response
Dear reviewer,
Thank you for your comments and suggestions during the review process. We found them extremely interesting and useful for improving our manuscript.
I hope that the changes we have made meet your requirements.
Comments 1: a method section describing the experimental proceduces, probes used etc. ist completely missing
Response 1: thank you for pointing out the probes; we have added the probes used in the FISH analysis. Regarding the methods section, we have specified all the experimental procedures in the various subsections: Cytogenetic analysis (line 105), Whole genome sequencing, and RNA sequencing analysis (line 162). We did not include a separate Materials and Methods section, but we would be glad to add one if you feel this would be more appropriate
Comments 2: labelling of figures is sometimes missing (e.g. Fig 1D), sometimes almost impossible to read (e.g. Fig 2C).
Response 2: we increased the image resolution to improve the labeling of the figures and labeled panel D more clearly in Fig. 1. We have also uploaded the images in a separate zip file.
Comments 3: discription of the rearrangement is confusing: while in line 107 a loss of 5q31 was reported, the rearrangement was overall described as balanced (line 95); it would be helpful to include the karyotype.
Response 3: thank you for your comment. The patient's karyotype is shown in panel A of Figure 1, highlighting the balanced translocation. We have clarified the description of the rearrangement, confirming the balanced translocation highlighted in the karyotype as indicated in lines 119 to 125.
Comments 4: do the authors have any explanation for the high VAF observed for ZRSR2 and KIT? any additional CNVs or CN-LOH that could explain that?
Response 4: thank you for this observation. We have decided to remove this information from the manuscript, considering that the variants in question did not play a diagnostic or prognostic role.
Comments 5: generally, although I agree that WGS and RNA seq was required to fully solve this case, it would not have been required for correct classification as myeloid/lymphoid neoplasm with PDGFRB rearrangement, FISH analysis would have been sufficient; this should at least be discussed and clarified what the benefit of WGS/RNA seq is
Response 5: thank you for your question. From line 199 to line 203, we indicate that WGS and RNAseq provided us with the fusion gene sequence, which was used to design primers and probes and monitor the patient during therapy.
Comments on the Quality of English Language: the manuscript is sometimes very wordy and explanations are confusing
According to the referees concern and comments, we have extensively edited the revised manoscript and double checked the text for typos and odd expressions. We hope that the manuscript, in its present form, is acceptable in terms of language and syntaxis used
I hope that the changes we have made meet your requirements. Every change made to the manuscript has been made in review mode so that you can evaluate the previous version and the revised version. We believe that the revised manuscript has been significantly improved by addressing all reviewers’ comments and we hope that it is now suitable for publication.
Best Regards
Cosimo Barbato

Reviewer 2 Report
Comments and Suggestions for Authors
The authors report the case of a 21-year-old patient with a myeloproliferative/ myelodysplastic neoplasm, presenting with hyperleukocytosis, anemia, thrombocytopenia, and elevated LDH.
Please delete the keyword- Case report.
In the introduction section, please include the incidence and the age at which it usually occurs.
The discussion section is too short and needs more text.
In high-risk patients with MDS/MPN and low blast count, what may be the best treatment?
Was there any organ infiltration, e.g., spleen? If there is splenomegaly, what may be the best treatment?
How can drug treatment vary according to the age of the patient?
Author Response
Comments 1: Please delete the keyword- Case report
Response 1: We agree, so we have removed the keyword “case report”
Comments 2: In the introduction section, please include the incidence and the age at which it usually occurs
Response 2: Thank you for this observation, which was previously missing. We have now included data on the incidence and age distribution (line 62-67)
Comments 3: The discussion section is too short and needs more text
Response 3: Thank you to emphasize this point. We have added information to the discussion section (line 222), including the conclusions, so that the section is not too short.I hope that the changes made are acceptable
Comments 4: In high-risk patients with MDS/MPN and low blast count, what may be the best treatment?
Response 4: We thank the reviewer for this comment. As outlined in the latest update of the EBMT 2025 guidelines in patients with MDS/MPN exhibiting proliferative features, such as leukocytosis, circulating immature myeloid cells, and/or splenomegaly, but with a low blast count, current recommendations generally support cytoreductive therapy, most commonly with oral hydroxyurea and hypomethylating agents. In the present case, hydroxyurea was administered (line 91-92) prior to the definitive diagnosis of a myeloid/lymphoid neoplasm with eosinophilia and tyrosine kinase gene fusion. Myeloid/lymphoid neoplasms with PDGFRB rearrangements are highly sensitive to tyrosine kinase inhibitors, particularly imatinib. Imatinib has been shown to induce rapid and durable hematologic and molecular remissions, even in patients with aggressive or proliferative features, provided the blast count remains low and no disease progression occurs. Therefore, although the patient had suspected high-risk MDS/MPN, the identification of a PDGFRB rearrangement guided the decision to initiate targeted therapy with imatinib. This approach aligns with current WHO and ICC classifications and published clinical evidence. We have added a brief discussion on this aspect in the revised manuscript (line 70-76)
Comments 5: Was there any organ infiltration, e.g., spleen? If there is splenomegaly, what may be the best treatment?
Response 5: Thank you for this observation. Diagnostic imaging revealed splenomegaly, and the precise volume of the spleen was included in the text as we believe this information to be useful (line 90-91). We apologize for not including it previously. In the present case, cytoreductive therapy with hydroxyurea as first-line treatment, followed by targeted therapy with imatinib, resulted in a significant reduction in spleen volume (line 253-256)
Comments 6: How can drug treatment vary according to the age of the patient?
Response 6: We acknowledge the reviewer’s point. Drug treatment may vary significantly according to patient age, as younger patients are more likely to tolerate long-term tyrosine kinase inhibitor therapy and benefit from sustained molecular monitoring aimed at maintaining deep remission. We have added this information to the text (line 241-244).
